# Adoption and implementation of teleaudiology as a telehealth model in Jordan and Arab countries: A cross-sectional survey

Hala M. AlOmari *, Hanady Bani Hani, Telda Alkhateeb, Dua' Qutaishat

Department of Hearing and Speech Sciences, University of Jordan, Amman, Jordan

* h_omari@ju.edu.jo

## Abstract

### Background and objectives

Telehealth is the provision of healthcare services remotely via telecommunications technology. The implementation, clinical applications, and perceived effectiveness of telehealth among audiologists across the Arab region, particularly following its accelerated adoption due to the COVID-19 pandemic was investigated.

### Materials and methods

A cross-sectional survey was conducted between April and June 2024 among 194 audiologists from multiple countries. A non-probability purposive sampling approach was implemented. The respondents were grouped into providers and non-providers of telehealth services. The questionnaire collected data on demographics, service delivery models, telehealth applications, training background, and perceived challenges. Descriptive and inferential analyses were performed to identify predictors of teleaudiology adoption.

### Results

46.9% of the sample reported providing telehealth services. Many of them (69.2%) indicated that they began offering telehealth services following the COVID-19 pandemic. Synchronous delivery was commonly utilised. Younger professionals and those employed in public institutions were more likely to engage in remote service delivery ($p < 0.05$). Barriers included limited formal training and limited infrastructure. Remarkably, 94% of non-providers expressed interest in implementing teleaudiology in the future.

### Conclusions

The audiologists' reported perceptions and experiences indicate that teleaudiology remains limited in clinical diagnostic service delivery. Broader integration of

**Data availability statement:** Data from all the experiments are available from Zenodo repository https://zenodo.org/records/17344611.

**Funding:** The author(s) received no specific funding for this work.

**Competing interests:** The authors have declared that no competing interests exist.

teleaudiology practices may benefit from enhanced professional training, the development of standardised guidelines, and investment in technological infrastructure to support access to remote hearing healthcare.

## Introduction

Telehealth, the provision of healthcare services remotely via telecommunications technology, has emerged as an innovative approach to addressing challenges in healthcare delivery and access [1,2]. Telehealth has been widely adopted among healthcare providers across various fields – including audiology, known as teleaudiology– as an alternative approach to providing and maintaining healthcare services that meet both providers' and patients' needs [3,4].

Teleaudiology uses telecommunications to provide audiological services remotely, encompassing a range of services, including hearing screenings [5], diagnostic evaluations [6], hearing aid fittings [7], and follow-up care, via synchronous (real-time) or asynchronous (store-and-forward) methods [1,8,9].

Prior to the pandemic, a scoping review of teleaudiology research [10]– based mainly on studies from high-income countries – indicated that teleaudiology demonstrated strong potential in expanding access to care, especially in underserved areas. This research has demonstrated the feasibility and reliability of remote audiological care on remote pure-tone audiometry, video otoscopy, and hearing aid programming [1,6–8], highlighting pathways for interdisciplinary collaboration and task-shifting through trained facilitators [10]. Moreover, studies have shown that both clients and practitioners report increased satisfaction and accessibility, especially among populations in rural or remote locations [11]. Teleaudiology is now recognised for its ability to mitigate geographical barriers, compensate for specialist shortages, and meet the rising demands of audiological services, especially in underserved areas [12,13].

Despite these promising developments, implementing teleaudiology is not without challenges. Commonly reported challenges include poor internet connectivity, limitations in performing certain diagnostic procedures remotely, and concerns regarding the quality of patient–provider interactions in a virtual setting [1,3,4,14]. Furthermore, many practitioners reported feeling unprepared for the sudden transition to telehealth, highlighting the need for more comprehensive training and education in this field [15]. Results from a study of perceptions of teleaudiology among Australian healthcare stakeholders [16] showed that, although most participants were familiar with teleaudiology and held positive views, client awareness and engagement were low. Key barriers included limited client offerings, low uptake in clinical placements, and insufficient support, indicating the need for greater understanding and collaboration to promote teleaudiology adoption.

Existing studies are largely derived from high-income healthcare systems, where technological infrastructure is robust, creating a knowledge gap regarding its implementation and the unique challenges faced in low- and middle-income countries [1,3], where healthcare infrastructure, digital readiness, regulatory frameworks, and access to audiological services vary considerably [1,3]. In this domain, a review of literature

on telepractice implementation in Africa revealed that practitioners face significant barriers related to infrastructure, training, and resource limitations. However, the potential of telepractice to expand access to care, particularly for vulnerable populations in remote and underserved areas, is profound [17].

Only a limited number of studies have examined audiologists' perceptions and attitudes toward teleaudiology in Arab countries [18,19]. One study included both audiologists and speech–language pathologists [20], while a few studies have focused on speech–language pathologists [21,22–24].

Specifically, Elbeltagy et al. [18] investigated attitudes toward teleaudiology among 112 audiologists in Egypt and Saudi Arabia during the COVID-19 pandemic. The results revealed that only 16% viewed teleaudiology positively, and 25.4% practised teleaudiology services mainly for counselling (73.3%), cochlear implant troubleshooting (46.7%), and hearing aid fitting (40%). Challenges reported by participants included the inability to provide full services, equipment shortages, and difficulties for patients in using technology. In addition, Zaitoun et al. [19] surveyed 164 audiologists in Jordan and other Arab countries; 61.6% were familiar with teleaudiology, and 48.2% offered it to patients. Most believed that telehealth will grow over the next decade (88.4%) and expressed a desire for further education (97.6%). They considered counselling and hearing aid follow-ups suitable for telehealth settings, but only a few believed that audiological assessments could be conducted remotely. Participants' concerns included patient privacy and payment claims. It should be noted that all these studies were conducted during the pandemic or shortly afterwards, with the focus on service delivery.

Despite the rapid global expansion of teleaudiology and substantial evidence supporting its effectiveness in improving access to hearing healthcare and patient outcomes, there remains a notable lack of region-specific evidence from Arab countries. The absence of systematic investigations into the readiness, perceived barriers, and enabling factors for teleaudiology adoption in Arab healthcare systems constitutes a critical gap in the literature. Addressing this gap is essential for informing context-appropriate implementation strategies and guiding the sustainable integration of teleaudiology into hearing healthcare services across Arab countries [3].

Accordingly, this study addresses the following research questions: (1) What types of audiology services are delivered remotely? (2) What are audiologists' attitudes toward teleaudiology? (3) What challenges are encountered in teleaudiology practice? and (4) What factors are associated with teleaudiology provision?

## Aims and objectives

By examining the services audiologists provide remotely, this study aims to provide a comprehensive understanding of the types of audiological care that can be effectively delivered through teleaudiology in the Arab region. In addition, identifying challenges in using teleaudiology will allow us to provide recommendations to improve teleaudiology services and make them more accessible and effective in the Arab region, as well as in other regions with similar healthcare infrastructure.

The few previous studies in this area have primarily focused on service delivery, but research on the perspectives of healthcare professionals regarding teleaudiology remains limited [14,25].

The objectives of this study are to:

• To examine the types of audiology services that are delivered remotely by practitioners.

• To understand audiologists' attitudes and perceptions of teleaudiology.

• To explore the challenges faced by audiologists when providing teleaudiology services.

• To gain insight into the perspectives of audiologists who do not currently offer teleaudiology services.

## Methods

A cross-sectional study was conducted in Arab countries between April and June 2024. The survey was distributed online to audiologists via email, WhatsApp, and social media platforms such as Facebook Messenger and LinkedIn. Participation

 

in this study was voluntary. Before proceeding to the survey, participants read a brief description of the study and the inclusion criteria. They then provided consent by checking a box to indicate their agreement to participate. Completing the survey took approximately 15–20 minutes. The study was approved by the Institutional Review Board/University of Jordan (Ref: 19/2023/470).

## Sample size

Information on the number of practising audiologists and technicians in Arab countries remains limited. Notably, only a limited number of Arab countries offer university programmes in audiology, with most being at the undergraduate level.

Jordan, where the majority of study participants are from, established the region's first undergraduate audiology programme in the early 1990s. Al Sabi [26] reported that there are 27 licensed audiologists in Jordan with postgraduate degrees. In addition, there are 244 audiology technicians and 7 assistant audiology technicians, all holding undergraduate degrees. These numbers are likely to have increased in the past few years.

These figures underscore the limited number of professionals with advanced qualifications in audiology within Jordan and Arab countries. The sample size of 100 audiologists in the current study is appropriate, as it is consistent with previous research evaluating similar outcomes among audiologists [18,19].

## Participants

A total of 194 audiologists participated in the online survey, with 103 (53.1%) identifying as non-providers of teleaudiology and 91 (46.9%) as teleaudiology providers. Participants spanned 22 countries, with the highest proportion of responses being from Jordan (37.6%), followed by Saudi Arabia (10%), Lebanon (9.3%), Kuwait (8.2%), Egypt (7.2%), and the United Arab Emirates (5.2%). The remaining responses were distributed across various Arab countries, each contributing less than 5%.

## The survey

The survey was adapted and curated based on a comprehensive review of previously published telehealth and teleaudiology instruments [14,21]. Items were mapped to established frameworks assessing telehealth readiness, clinician attitudes, service feasibility, perceived barriers, and satisfaction. Content and face validity were established through expert review by five senior audiologists with experience in telepractice, ensuring relevance, clarity, and contextual appropriateness for the Arab countries.

While the focus of the study was Audiologists who used teleaudiology (section 2) the survey also included questions for audiologists who did not have experience with teleaudiology to explore their perception of teleaudiology (section 3).

The survey consisted of four main sections. The first section, 'Demographic Information', included 12 questions. The questions gathered data on age, gender, profession, level of education, country of employment, work setting, years of work experience, types of clients served, and services provided.

The second section, comprising ten main domains, was directed at audiologists who provided teleaudiology services. These domains were: modality and platforms used, training, practice, and services; services that worked the best; services that worked the least; challenges faced while providing services; benefits, barriers; clients' main complaints; and the cost of teleaudiology.

The third section was directed at audiologists who did not provide teleaudiology services (non-providers). This section consisted of four main domains: clinical competencies required for teleaudiology; demands of teleaudiology; barriers to providing teleaudiology.

The fourth section was 'Attitudes Regarding Teleaudiology'. It comprised five questions designed to explore audiologists' perceptions of and attitudes towards teleaudiology. These questions were adapted from Nihara et al.'s study [27].

The majority of questions were multiple-choice, with some following a 'Yes/No' format. For multiple-choice questions, respondents were allowed to select more than one response, where applicable.

## Statistical analysis

Data were exported to SPSS version 29.0 for analysis. Descriptive statistics (frequencies, means, and standard deviations) were computed to summarize participant characteristics and responses. Chi-square and independent-sample t-tests were used to examine associations between categorical and continuous variables, respectively. Binary logistic regression analysis was performed to identify predictors of teleaudiology adoption. Statistical significance was set at $p < 0.05$. Analyses examining associations between demographic, professional variables and teleaudiology provision were exploratory.

## Results

### Providers of teleaudiology

**Characteristics and professional information.** Most of the 91 respondents in the teleaudiology provider group were female and aged 21–35. They all held a degree in audiology (BSc: 58.2%, MSc: 11%, PhD &AuD: 23.1% and Medical degree: 7.7%), as illustrated in Table 1.

The respondents were mostly from Jordan (31.9%), followed by Saudi Arabia (15.4%) and Lebanon (12.1%). The remaining respondents were distributed across other Arab countries.

As shown in Table 2, most participants (58.2%) had less than 4 years of experience and worked in various settings, with 53.3% employed by private clinics. The respondents provided services for both children (81.3%) and adults (73.6%), with the most common practices including hearing aid fitting (73.6%), paediatric hearing evaluations (67%), and adult hearing evaluations (60.4%).

Table 1. Personal and educational characteristics of teleaudiology providers and non-providers.

| Area | Teleaudiology users (*n*=91) | | Non-teleaudiology users (*n*=103) | |
|---|---|---|---|---|
| **Gender** | **Number** | **Percentage** | **Number** | **Percentage** |
| Male | 35 | 38.5 | 26 | 25.2 |
| Female | 56 | 61.5 | 77 | 74.8 |
| **Age group** | **Number** | **Percentage** | **Number** | **Percentage** |
| < 25 years | 24 | 26.4 | 38 | 36.9 |
| 25-30 years | 28 | 30.8 | 37 | 35.9 |
| 31–35 years | 17 | 18.7 | 10 | 9.7 |
| 36–40 years | 3 | 3.3 | 10 | 9.7 |
| 41–45 years | 7 | 7.7 | 1 | 1 |
| 46-50 years | 6 | 6.6 | 3 | 2.9 |
| 51-55 years | 2 | 2.2 | 3 | 2.9 |
| > 56 years | 4 | 4.4 | 1 | 1 |
| **Education** | **Number** | **Percentage** | **Number** | **Percentage** |
| Bachelor degree (BSc.) | 53 | 58.2 | 70 | 68.0 |
| Masters degree (MSc.) | 10 | 11.0 | 7 | 6.8 |
| Doctor of philosophy (PhD) | 12 | 13.2 | 3 | 2.9 |
| Doctor of Audiology (AuD) | 9 | 9.9 | 18 | 17.5 |
| Medical Audiology | 7 | 7.7 | 5 | 4.9 |

\* Some Audiologists work in more than one location.

\*\*The percentage was calculated based on the total number of respondents in each group.

**Table 2. Professional practice-related characteristics of teleaudiology providers and non-providers.**

| Area | Teleaudiology users (*n*=91) | | Non-teleaudiology users (*n*=103) | |
|---|---|---|---|---|
| **Profession** | **Number** | **Percentage** | **Number** | **Percentage** |
| Audiologist | 75 | 82.4 | 92 | 89.3 |
| Medical Audiologist | 16 | 16.6 | 11 | 10.7 |
| **Work experience** | **Number** | **Percentage** | **Number** | **Percentage** |
| 0-4 years | 53 | 58.2 | 72 | 69.9 |
| 5-9 years | 27 | 29.7 | 22 | 21.4 |
| > 10 years | 11 | 12.1 | 9 | 8.7 |
| **Clients** | **Number*** | **Percentage**** | **Number*** | **Percentage**** |
| Children | 74 | 81.3 | 84 | 81.6 |
| Young adults | 57 | 62.6 | 56 | 54.4 |
| Adults | 67 | 73.4 | 63 | 61.2 |
| Elderly | 50 | 54.9 | 50 | 48.5 |
| **Area of practice** | **Number*** | **Percentage**** | **Number*** | **Percentage**** |
| Paediatric hearing evaluation | 61 | 67.0 | 73 | 70.9 |
| Adult hearing evaluation | 55 | 60.4 | 62 | 60.2 |
| Hearing aid fitting | 67 | 73.6 | 62 | 60.2 |
| Cochlear implants | 41 | 45.1 | 29 | 28.2 |
| Vestibular testing and evaluation | 14 | 15.4 | 14 | 13.6 |
| Tinnitus testing and rehabilitation | 24 | 26.6 | 19 | 18.4 |
| **Work setting** | **Number*** | **Percentage**** | **Number*** | **Percentage**** |
| Self employed | 12 | 11.1 | 10 | 10.2 |
| Government employee | 8 | 8.9 | 21 | 21.4 |
| Public sector | 14 | 15.6 | 12 | 12.2 |
| Private sector | 48 | 53.3 | 44 | 44.9 |
| School setting | 3 | 3.3 | 3 | 3.1 |
| Hospital setting | 18 | 20.0 | 18 | 18.4 |

* Some Audiologists work in more than one location.

**The percentage was calculated based on the total number of respondents in each group.

Moreover, 76.9% of respondents reported receiving formal training in teleaudiology; 16.5% indicated they were self-taught through self-help guides, video conferencing, and articles; and 6.6% reported having received no training. Of those who reported receiving training, 57.1% of the respondents completed it through online courses or lectures, 26.4% through websites, and 25.3% through workplace workshops. In addition, 19.8% of the respondents received their training before COVID-19, 7.7% during the lockdown, and 57.1% post-pandemic.

**Attitudes towards teleaudiology.** Teleaudiology providers were asked to provide their opinions on statements regarding teleaudiology, as detailed in Table 3.

The internal consistency of the five-item attitude scale for teleaudiology was assessed using Cronbach's alpha, which yielded a value of approximately 0.70, indicating acceptable reliability. No single item significantly increased this value if deleted, suggesting that all items contribute meaningfully to the overall construct.

Regarding the difference in interaction quality between teleaudiology and in-person services, most responses indicated neutral view (48.4%). In contrast, there was strong agreement that teleaudiology expands geographical reach (strongly

Table 3. Attitudes of non-providers (N) and providers (P) of teleaudiology services.

| Statement | | Strongly Agree | Agree | Neutral | Disagree | Strongly Disagree |
|---|---|---|---|---|---|---|
| No difference in interaction quality | N: 9 (8.7%) | 30 (29.1%) | 38 (36.9%) | 24 (23.3%) | 2 (1.9%) |
| | P: 9 (9.9%) | 24 (26.4%) | 44 (48.4%) | 9 (9.9%) | 5 (5.5%) |
| Teleaudiology expands geographical reach | N:19(18.4%) | 63 (61.2%) | 17 (16.5%) | 4 (3.9%) | 0 (0.0%) |
| | P: 26 (28.6%) | 39 (42.9%) | 21 (23.1%) | 3 (3.3%) | 2 (2.2%) |
| Teleaudiology has promising commercial prospects | N: 11 (10.7%) | 64 (62.1%) | 24 (23.3%) | 4 (3.9%) | 0 (0.0%) |
| | P: 18 (19.8%) | 47 (51.6%) | 20 (22.0%) | 4 (4.4%) | 2 (2.2%) |
| Teleaudiology requires more effort | N: 17 (16.5%) | 66 (64.1%) | 17 (16.5%) | 2 (1.9%) | 1 (1.0%) |
| | P: 21 (23.1%) | 43 (47.3%) | 22 (24.2%) | 1 (1.1%) | 4 (4.4%) |
| Teleaudiology will replace face-to-face | N:11 (10.7%) | 41 (39.8%) | 21 (20.4%) | 23 (22.3%) | 7 (6.8%) |
| | P:15 (16.5%) | 36 (39.6%) | 24 (26.4%) | 8 (8.8%) | 8 (8.8%) |

N = Non providers

P = Providers.

agree: 28.6%; agree: 42.9%) and its commercial potential (strongly agree: 19.8%; agree: 51.6%). Most respondents also agreed that teleaudiology requires more effort compared with face-to-face services (strongly agree: 23.1%; agree: 47.3%). Finally, opinions were more divided regarding whether teleaudiology would replace traditional in-person care, as shown in Table 3.

**Services and practice.** Results showed that 49.5% of teleaudiology providers were satisfied with teleaudiology services, 47.3% preferred a combination of in-person and teleaudiology, and 3.3% were dissatisfied.

The majority of teleaudiology providers (69.2%) indicated that they began offering teleaudiology services after the COVID-19 pandemic and have continued to do so. Meanwhile, 11% of respondents began providing these services during the COVID-19 lockdown, and 19.8% had already been offering teleaudiology services prior to the pandemic, as shown in Fig 1. Among those who offered teleaudiology services pre-pandemic, 2 of 18 respondents reported discontinuing the service. The main reasons cited for discontinuation were lack of knowledge, insufficient electronic equipment, the absence of suitable online applications, and the high costs of teleaudiology services. In addition, 2 of 10 respondents who began offering teleaudiology during lockdown stopped offering these services afterwards. Their main reasons for discontinuation were patients' preference for in-person sessions and the challenges posed by online audiology sessions.

Technical/device-based clinical services, such as hearing aid programming and adjustment, followed by first-time fittings for experienced hearing aid users and cochlear implant mapping, were among services often offered through teleaudiology. While diagnostic and screening services were least offered through teleaudiology, as shown in Fig 2.

Synchronous (real-time) teleaudiology use varied among respondents: 49.5% always or often used it, 47.3% sometimes used it, and 3.3% never used it. Asynchronous teleaudiology was less commonly utilised: 34.1% of respondents always or often used it, 52.7% sometimes used it, and 13.2% never used it. A hybrid approach combining synchronous and asynchronous methods was used always or often by 44% of respondents, sometimes by 44%, and never by 12.1%. Additionally, 41.8% of the respondents always or often worked with an assisting audiologist, 36.2% sometimes did so, and 22% never did.

Computers were the primary modality for delivering teleaudiology, used by 87.9% of the respondents, followed by tablets (35.2%), smartphones (33%), and telephones (22%).

FaceTime was the most frequently used platform for teleaudiology services (68.9%), followed by telephone calls (42.2%). The remaining responses cited platforms such as WhatsApp, Zoom, Microsoft Teams, Facebook Messenger, and other specialised teleaudiology platforms (15.6%).

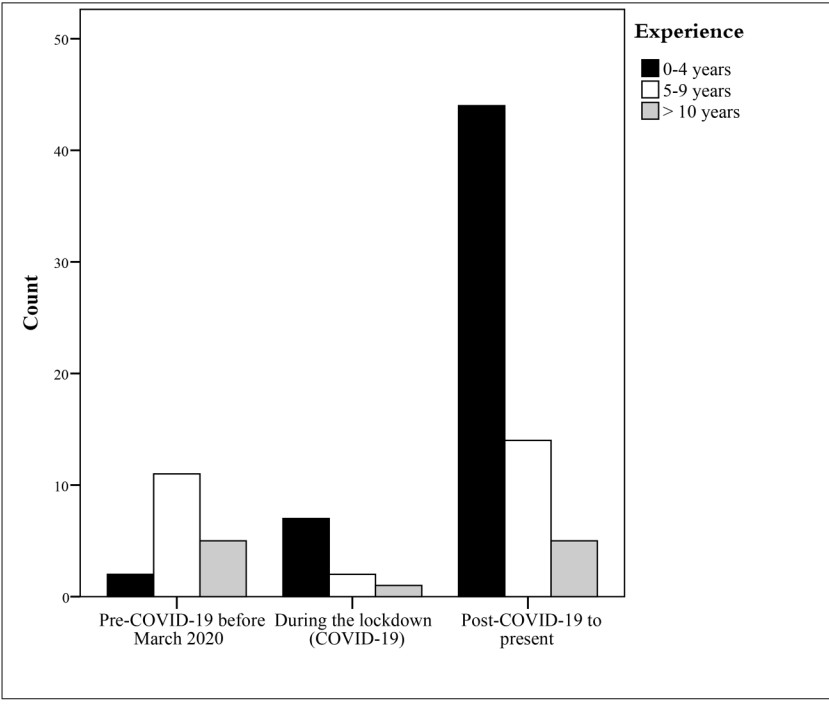

**Fig 1. Teleaudiology providers grouped by time of teleaudiology adoption and years of experience.**

Most of the respondents (60.4%) believed that the cost of a teleaudiology session should be the same as an in-person session. Meanwhile, 20.9% felt it should be more expensive, whereas 18.7% indicated it should be less expensive.

**Perception of the most effective vs least effective services.** Fig 3 compares participants' perceptions of the most and least effective services in teleaudiology settings. Programming adjustment emerged as the most successful service, with the highest number of 'best' ratings. This was followed closely by experienced fitting, first-time fitting, and cochlear implant mapping.

Conversely, troubleshooting was most frequently cited as the least effective service. Other services with notable 'least effective' ratings included aural rehabilitation, screening, client training, and counselling.

**Challenges and barriers faced by practitioners offering teleaudiology.** Teleaudiology providers reported several challenges, including a lack of knowledge of teleaudiology (41.1%); limited patient access to computers, tablets, and video equipment (37.4%); poor internet access (35.2%); and difficulties with the equipment required for remote assessments (23.1%).

**Clients' main complaints.** Respondents were asked to identify the most common client complaints. They reported experiencing difficulties on the client side relating to technology, including connectivity issues (87.9%), environmental distractions (85.7%), and limited knowledge of how to use equipment and computers (58.2%). Additionally, 47.3% reported that clients preferred in-person interactions, and 37.4% noted that patients found teleaudiology sessions time-consuming and requiring significant preparation. Furthermore, 33 participants (36.3%) mentioned that teleaudiology increased the demands of their role, 23 (25.3%) cited slow internet speed as a concern, and 9 (9.9%) noted that patients often worry about confidentiality.

## Non-providers

**Characteristics and professional information.** Among respondents in the non-provider group, many of whom had no prior teleaudiology experience, the majority held a degree in audiology (BSc: 68.0%, MSc: 6.8%, PhD & AuD: 20.4%,

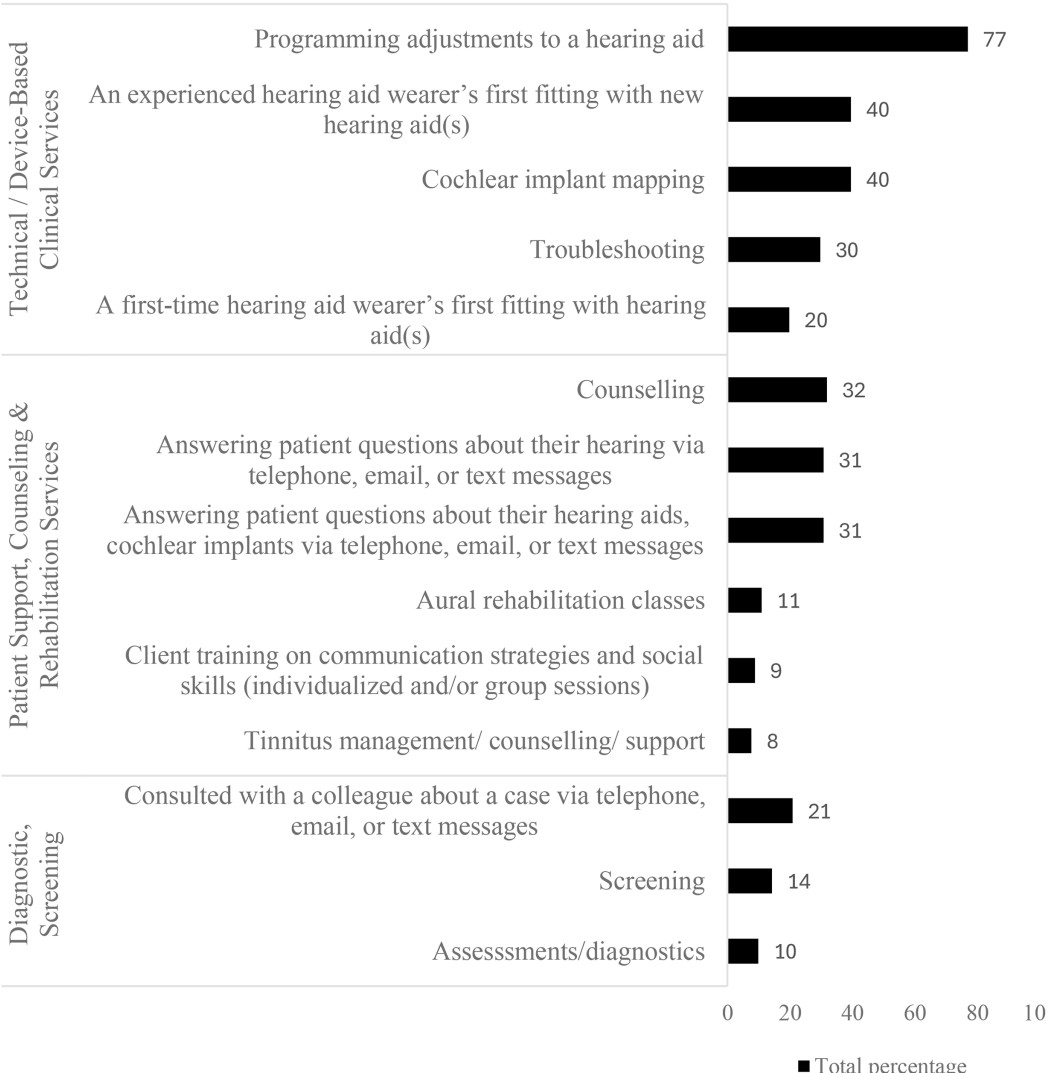

**Fig 2. Clinical services provided via teleaudiology.**

and Medical degree: 4.9%). Most of the respondents were women (74.8%) and younger than 30 years (42.7%) from Jordan. They worked in different settings, with 40.7% in private practice and 19.4% in government. In addition, 69.9% had less than 4 years of experience.

Tables 1 and 2 display the personal and professional characteristics of the non-provider group. Most respondents (81.6%) reported working with children, (61.2%) with adults, (54.4%) with young adults, and (48.5%) with older adults. The most common services provided included paediatric hearing evaluation (70.9%), adult hearing evaluation (60.2%), hearing aid fitting (60.2%), cochlear implant mapping (28.2%), tinnitus evaluation and management (18.4%), and vestibular evaluation (13.6%).

**Teleaudiology perception.** Non-providers of teleaudiology services identified key clinical competencies essential for effective service delivery from their perspectives. The most frequently reported competency reported by 57.3%

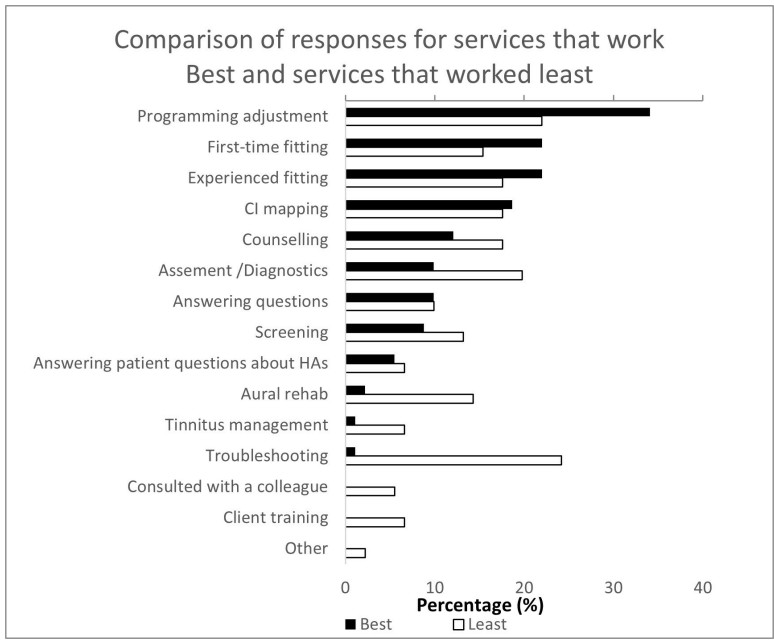

**Fig 3. Comparison of participants' perceptions of teleaudiology services rated as most effective and least effective.**

of the respondents were the ability to maintain client engagement and use interactive equipment and applications without service disruption. This was followed by ensuring a distraction-free virtual setting, as reported by 54.4% of respondents. Maintaining eye contact with remote clients was also emphasised, with 53.4% respondents highlighting its importance.

**Perceived barriers.** The perceptions of non-providers of teleaudiology services. There was agreement among non-providers across the different domains that teleaudiology services are more demanding than in-person services. Specifically in terms of technical support (87.4%), the involvement of caregivers or helpers in the preparation of sessions (79.6%), reinforcement (77.8%), session preparation time (75.7%), and therapy materials preparation (74.8%).

Moreover, respondents reported several barriers to offering teleaudiology services, with the most common being a lack of knowledge (47.1%), followed closely by client preference for in-person sessions (45.1%), a lack of suitable online applications (43.1%), insufficient electronic equipment for therapy (34.3%), and challenges related to the audiology session environment (34.3%).

Despite these barriers, the majority (94.2%) expressed a desire to expand their services to include teleaudiology in the future.

**Attitudes towards teleaudiology.** Non-providers of teleaudiology were asked to provide their opinions on statements regarding teleaudiology, as detailed in Table 3. Regarding the difference in interaction quality between teleaudiology and in-person services, most responses were neutral (36.9%) or disagreed (25.2%). In contrast, there was strong agreement that teleaudiology expands geographical reach (strongly agree: 18.4%; agree: 61.5%) and its commercial potential (strongly agree: 10.7%; agree: 62.1%). Most respondents also agreed that teleaudiology requires more effort compared with face-to-face services (strongly agree: 16.5%; agree: 64.1%). Finally, opinions were more evenly divided on whether teleaudiology would replace traditional in-person care, with half of the respondents indicating agreement (strongly agree: 10.7%; agree: 39.8%).

### Providers vs non-providers

**General comparison.** An assessment of the differences between providers and non-providers is shown in Table 4. Providers of teleaudiology services were generally younger, had slightly higher rates of postgraduate education, and were more likely to work in private clinics than non-providers. They also had more formal training in teleaudiology, especially post-pandemic, and offered advanced services such as hearing aid programming and cochlear implant mapping. In contrast, non-providers lacked formal training, focused more on basic evaluations and fittings, and cited barriers such as limited knowledge, equipment constraints, and client preference for in-person care.

A logistic regression analysis was conducted to examine the influence of demographic and professional factors on the likelihood of providing telehealth services ('Yes' vs 'No'). Age category was a highly significant predictor ($p < 0.001$), with negative regression coefficients indicating that higher age categories were associated with significantly lower odds of providing telehealth services. Profession, education, and gender were not statistically significant predictors of telehealth provision ($p > 0.05$).

Mann–Whitney U tests were conducted to examine differences in clinical practice areas between professionals who provide telehealth services and those who do not. The results indicated statistically significant differences in two areas: hearing aid fitting ($U = 4,057.00$, $Z = -1.973$, $p = 0.049$) and cochlear implant services ($U = 3,894.50$, $Z = -2.440$, $p = 0.015$). This suggests that professionals who offer these practices are less likely to provide telehealth services.

Logistic regression was performed to assess the relationship between work setting and the likelihood of providing telehealth services. None of the work-setting variables were statistically significant predictors (all $p > 0.05$), suggesting no clear association between work setting and telehealth provision in this sample.

## Discussion

This is the first study to explore audiologists' perceptions of teleaudiology in Jordan and the Arab region following the COVID-19 pandemic. A total of 194 audiologists participated in the online survey, with 103 identifying as non-providers of teleaudiology and 91 as teleaudiology providers. Participants came from 22 countries, with the largest share of responses from Jordan

**Table 4. Comparison of providers and non-providers of teleaudiology.**

| Aspect | Providers | Non-Providers |
|---|---|---|
| Gender Distribution | 61.5% Female, 38.5% Male | 74.8% Female, 25.2% Male |
| Country Representation | Mostly from Jordan, Saudi Arabia, Lebanon | Majority from Jordan, Egypt, Kuwait |
| Age Group | 57.2% under 30 | 36.9% under 30 |
| Experience Level | 58.2% have <4 years experience | 69.9% have <4 years experience |
| Education | 58.2% Bachelor's, 35.1% Postgrad | 68% Bachelor's, 27.2% Postgrad |
| Work Setting | 53.3% Private clinics | 40.7% Private clinics |
| Client Groups Treated | Mostly children (81.3%), adults (73.6%) | Mostly children (81.6%), adults (61.2%) |
| Common Services | Hearing aid programming, cochlear implant mapping, troubleshooting | Paediatric/adult hearing evaluations, hearing aid fittings |
| Formal Training in Teleaudiology | 76.9% trained (57.1% post-pandemic) | Most lacked training |
| Barriers Faced | Lack of client technology, distractions, and technology knowledge gaps | Lack of knowledge, client preference for in-person, tech & equipment constraints |
| Client Complaints | Difficulty using technology, preference for in-person, session prep effort | Not directly reported, but inferred from cited barriers |
| Future Interest in Teleaudiology | Currently active | 94.2% interested in offering services in the future |

(37.6%), Saudi Arabia (10), Lebanon (9.3%), Kuwait (8.2%), Egypt (7.2%), and the United Arab Emirates (5.2%). The remaining responses were distributed across various Arab countries, each contributing less than 5%. Recent statistics regarding the number of audiologists in such countries are not currently available. The imbalance among countries is due to the lack of audiology educational programs in many Arab countries, with Jordan having the oldest undergraduate program in the region.

## Service delivery

The COVID-19 pandemic was a significant driver for the adoption of teleaudiology among audiologists in Arab countries. This was evident in the fact that most respondents (69.2%) reported offering teleaudiology services after the pandemic, reflecting global trends observed across healthcare sectors during and after the lockdown period [28]. However, the reactive nature of this adoption is evident in the fact that more than half of those trained (57.1%) received training only after the pandemic, indicating a lack of preparedness and infrastructure prior to the global health crisis. [29,30] Although teleaudiology services and relevant regulations were established in the developed world in the early 2000s [29,30], teleaudiology use in the Arab region was limited to hearing aid programming and troubleshooting; however, remote diagnostic testing was not utilised. This is consistent with the recommendations of Coco et al. [31]that audiology services require minimal use of technology, such as hearing aid consultations, troubleshooting, and aural rehabilitation can be used remotely.

Providers of teleaudiology services offered them via a synchronous (real-time) (49.5%) approach or a hybrid approach (44%). The modality of delivery was via computers (68.9%), primarily via platforms such as FaceTime, followed by telephone calls. These results are similar to those reported in previous studies [11,32].

Overall findings suggest that device-related adjustments and technical support are the most common services offered via teleaudiology, whereas rehabilitation, training, and diagnostic assessments are less frequently conducted remotely. This highlights a significant challenge in providing remote support for technical or device-related issues.

Our findings indicate that younger practitioners are more eager to apply technological advancements. In contrast, Elbeltagy et al. [18] reported a statistically significant association between teleaudiology services and providers aged 40–49 years. Meanwhile, other studies found no significant association between teleaudiology practice and participant age [33]. These comparisons suggest that while our results indicate that younger practitioners have a higher tendency to adopt technological advancements, the relationship between age and the likelihood of offering teleaudiology services remains inconsistent across the literature.

A total of 55 respondents (60.4%) believed that the cost of a teleaudiology session should be the same as an in-person session. Meanwhile, 19 respondents (20.9%) felt it should be more expensive, and 17 (18.7%) thought it should be less expensive. The American Speech-Language-Hearing Association guidelines specify that regulations for teleaudiology services are the same as those for in-person services. However, travel costs are lower for remote services [34].

Just over half of respondents (56%) believed that teleaudiology improved accessibility and lowered costs compared with traditional services. Other notable benefits cited included increased convenience, improved access to therapy otherwise unavailable due to distance, greater caregiver involvement, and improved therapy regularity. These findings echo prior work, such as Blyth and Saunders's research [35], which found that patients were generally satisfied with teleaudiology for hearing aid support but still preferred in-person fittings for long-term use. Healthcare practitioners appreciated the convenience and flexibility of teleaudiology services, but they raised concerns regarding communication and rapport. Similarly, Binkhamis et al. [20] reported that while 68.4% of clinicians in Saudi Arabia had experience in telehealth, many found telehealth more effective for consultation and counselling than for direct intervention, and attitudes towards telehealth were largely positive among those familiar with it.

## Challenges and barriers to teleaudiology

**Technological barriers.** One of the most prominent barriers to effective teleaudiology is the lack of technological infrastructure. Among teleaudiology providers, 37.4% reported limited patient access to devices such as computers,

tablets, and video equipment, while 35.2% cited poor internet connectivity as a significant issue. Additionally, 23.1% reported difficulties with the equipment needed to conduct remote assessments. These technological limitations not only affect clinicians but also restrict patients' ability to participate in teleaudiology sessions, reflecting the digital divide observed across healthcare fields [1,4,11,14,36].

**Client-related barriers.** From the respondents' perspectives, client-side challenges in teleaudiology were particularly pronounced, with the majority highlighting technological and environmental barriers. A striking 87.9% reported that clients lacked adequate technology – a well-documented issue in the literature, especially among older adults and those in underserved regions where the digital divide remains a critical obstacle [16,35,37,38]. Environmental distractions were cited by 85.7% of respondents, echoing findings that home settings often compromise the clinical integrity of remote sessions [31,37]. Digital literacy emerged as another major concern, with 58.2% noting limited ability to use electronic devices among patients and 90.1% agreeing that basic technological skills are essential for successful engagement – consistent with broader research emphasising the need for user-friendly platforms and pre-session support [16,35,37,38]. Preferences for in-person care were also evident, with 47.3% of respondents indicating that clients favoured face-to-face interactions, often due to comfort, familiarity, or perceived quality [4,35]. Additionally, 37.4% noted that clients found teleaudiology sessions time-consuming or demanding in terms of preparation – a sentiment that varied across the study population but reflects concerns regarding workflow and accessibility [35,37]. Although less frequently mentioned, concerns regarding confidentiality were also reported (9.9%) – an issue that, while minor in prevalence, remains important for building trust in remote care models [38]. Collectively, these findings underline the need for targeted interventions that address technological access, digital literacy, and client preferences to enhance the effectiveness and equity of teleaudiology services.

**Practitioner perspectives.** In addition to technological and client-related challenges, practitioner attitudes and experiences also significantly shape the current landscape. Among non-providers (n = 103), 47.1% cited lack of knowledge as the primary barrier to offering teleaudiology services. This has also been frequently cited by non-providers in previous studies [11,16,37,38]. The second-most common challenge was client preference for in-person sessions, reported by 45.1%, which was also prevalent in prior studies [4,16]. Technological limitations associated with providing teleaudiology services were also frequently cited; diagnostic testing often requires specific equipment, leading many audiologists to believe that these services cannot be performed remotely. Furthermore, 43.1% referenced the absence of suitable online applications, while 34.3% cited insufficient electronic equipment for therapy, which also echoes the findings reported in previous studies [4,11,37,38]. Furthermore, practitioners often noted that teleaudiology requires more preparation time, additional therapy materials, and greater caregiver involvement [11,16,38], which is similar to the findings reported by non-providers of teleaudiology in the current study

## Perceptions of teleaudiology

The current study examined the attitudes of both teleaudiology providers and non-providers. Consistent with the findings of Nihara and Seethapathy [27], the results revealed that audiologists generally agreed that teleaudiology enhances geographic reach and holds promising commercial potential. Responses regarding the quality of online interactions were mixed, with a significant proportion of both providers and non-providers taking a neutral stance, which, again, aligns with the observations reported by Nihara and Seethapathy [27]. Additionally, Nihara and Seethapathy [27] found that almost 90% of both providers and non-providers believed that teleaudiology services often demand more effort than in-person services. This trend was similarly reflected in the current study, where most participants agreed that teleaudiology requires greater effort.

## Training and satisfaction with teleaudiology

Notable gaps in formal training for teleaudiology were observed among the participants. While 76.9% reported having received some form of training, this was not universal: 16.5% were self-taught, and 6.6% had received no training at all.

This disparity highlights the absence of standardised curricula and institutional support within the field, which may influence both the quality of teleaudiology services and clinicians' confidence. The reliance on online courses and workplace workshops underlines the importance of structured, evidence-based educational opportunities. These findings are consistent with those of Ravi et al. [38], who noted that approximately 90% of audiologists already practising teleaudiology expressed interest in increasing their knowledge through further education. Similarly, 75% of audiologists who had not yet adopted teleaudiology were interested in additional training – a trend echoed in the current study, as 94% of non-providers indicated a desire to expand their practice to include teleaudiology services.

In addition to training disparities, participant satisfaction with teleaudiology services offers further insight into current challenges and opportunities. In our study, 49.5% of respondents expressed satisfaction with teleaudiology services following the COVID-19 pandemic. Meanwhile, 47.3% preferred a hybrid approach combining in-person and teleaudiology services, and 3.3% reported dissatisfaction (with one participant providing teleaudiology only during lockdown and two providing it post-lockdown). For comparison, Elbeltagy et al. [18] reported a satisfaction rate of just 16% among practitioners during the pandemic. The increase observed in our study suggests that as practitioners have adapted to remote care following the pandemic, acceptance of and satisfaction with teleaudiology services have improved.

Overall, these findings illustrate that while teleaudiology offers clear benefits in accessibility, convenience, and expanded service reach, its widespread adoption remains hindered by gaps in formal training, technological limitations, and client-side digital literacy issues. Addressing these barriers through targeted, evidence-based training programmes, investment in infrastructure, and client support initiatives will be essential to unlocking the full potential of teleaudiology, both in the Arab region and globally. By focusing on the needs of clinicians and clients, and by bridging current divides, teleaudiology can become a sustainable and equitable solution for hearing healthcare delivery.

### Study limitations and future directions

The field of audiology relies on specialised equipment for the diagnosis and management of hearing loss, and many audiologists believe it is not feasible to provide these services remotely. This survey reached audiologists in 22 countries – more than any previously published study in the region. However, as with all surveys, there are limitations in the sample's reach and representativeness, which can lead to response bias favouring individuals familiar with digital platforms. Additionally, client perspectives were not assessed in this study. As such, future research should explore teleaudiology from the client perspective.

As the field continues to evolve, there is an urgent need for research into best practices, policy guidelines, and programme development to support the effective provision of teleaudiology services. Key priorities include assessing long-term outcomes, patient and provider satisfaction, and strategies to overcome current technical and procedural limitations.

### Conclusion

A promising surge in teleaudiology services was observed among audiologists in the Arab region, indicating their ability to align with global trends. Services offered through teleaudiology included device programming (hearing aids and cochlear implants) and counselling. However, various challenges and barriers were noted, particularly the lack of formal training. Gaps in professional preparation were identified, with programming adjustments for hearing aids, cochlear implant mapping, and first-time fittings for experienced users being the most frequently reported services.

It is recommended that Arab countries develop structured teleaudiology training programs that can be combined into educational curricula. Promote hybrid models that combine in-person and remote care. Investment in basic technological infrastructure.

## Institutional review board statement

The study was conducted in accordance with the Declaration of Helsinki and approved by the Institutional Review Board of University of Jordan (Ref: 19/2023/470 Date: 1/8/2023).

## Informed consent statement

The survey was disseminated online with audiologists via email, WhatsApp, and social media platforms such as Facebook Messenger, LinkedIn. Participation in this study was voluntary. Before proceeding to the survey, participants read a brief description of the study and the inclusion criteria, then provided consent by checking a box to indicate their agreement to participate.

## Acknowledgments

The authors express their gratitude to all the audiologists and colleagues who took the time to answer the survey. We would like to acknowledge Ms. Alissar Al-zoubi for her help in disseminating the survey.

## Author contributions

**Conceptualization:** Hala AlOmari.

**Data curation:** Hala AlOmari, Hanady Bani Hani, Telda Alkhateeb.

**Formal analysis:** Hala AlOmari.

**Methodology:** Hala AlOmari, Telda Alkhateeb.

**Project administration:** Hala AlOmari.

**Resources:** Dua' Qutaishat.

**Writing – original draft:** Hala AlOmari, Hanady Bani Hani, Telda Alkhateeb, Dua' Qutaishat.

**Writing – review & editing:** Hala AlOmari, Hanady Bani Hani, Dua' Qutaishat.

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
