## [Decision Letter · Decision Letter 0]

29 Dec 2025

Dear Dr. AlOmari,

Thank you for submitting your manuscript to PLOS ONE. After careful consideration, we feel that it has merit but does not fully meet PLOS ONE’s publication criteria as it currently stands. Therefore, we invite you to submit a revised version of the manuscript that addresses the points raised during the review process.

**ACADEMIC EDITOR:**

We look forward to receiving your revised manuscript.

Kind regards,

Rohit Ravi, Ph.D.

Academic Editor

PLOS One

Journal Requirements:

Reviewers' comments:

Reviewer's Responses to Questions

**Comments to the Author**

1. Is the manuscript technically sound, and do the data support the conclusions?

Reviewer #1: Partly

Reviewer #2: Yes

2. Has the statistical analysis been performed appropriately and rigorously?

Reviewer #1: Yes

Reviewer #2: Yes

3. Have the authors made all data underlying the findings in their manuscript fully available?

Reviewer #1: No

Reviewer #2: Yes

4. Is the manuscript presented in an intelligible fashion and written in standard English?

Reviewer #1: Yes

Reviewer #2: Yes

Reviewer #1: The paper could have had very valid information if published a year or two ago. There is currently sufficient evidence on telepractice-related issues. Many are using it as a routine. Since the authors have focused on the arab region, it can still be considered. Though the author declared data available, I could not get access. The following points need attention

1. Aims need to be rewritten. The aims listed sound more like objectives rather than aims

2. It is good to give stats on which countries and their percentage rather than just mentioning two countries and others are less than 10%. Because if there are only one or two from other countries, it is good to exclude those data and re-title to the region focused.

3. The outcomes of the study highlighted, though not unique, but author can highlight with emphasis to the region, focusing on what aspects can be better.

Reviewer #2: Reviewers comment:

Abstract:

• The abstract is structured and clearly reports background, methods, key results, and conclusions, allowing readers to quickly understand the scope of the study.

• Temper the conclusion by explicitly stating that findings reflect perceptions rather than effectiveness or outcomes.

Introduction

• The introduction provides a comprehensive overview of teleaudiology, supported by relevant and up-to-date international literature.

• The introduction is overly long and reads more like a narrative review than a focused rationale for the present study.

• Repetition is evident when discussing COVID-19–related acceleration of telehealth adoption. Methodological limitations of previous studies are not sufficiently highlighted. Add critical comparisons between studies rather than descriptive summaries.

• The transition from global literature to the Arab-region context is gradual and lacks a strong problem statement. Add a concise paragraph clearly identifying the research gap and problem statement.

• Research questions or hypotheses are not explicitly articulated.

Methods

• Ethical approval, participant consent, and statistical procedures are clearly documented, enhancing transparency and ethical rigor.

• No validation process for the adapted questionnaire is described beyond internal consistency. Describe content or face validity procedures for the survey also the content validity Index (CVI) for the questionnaire used.

• The rationale for grouping providers vs. non-providers is not explained in the study.

• Country-wise sample imbalance (e.g., dominance of Jordan) is not addressed.

• Potential response bias due to online distribution of the questionnaire is not discussed.

Results

• Results are detailed, systematically presented, and supported by tables and figures that enhance clarity.

• The results section is excessively long and contains interpretative statements better suited for the discussion.

• Some statistical findings (e.g., extremely small odds ratios) are reported without sufficient explanation.

• Multiple analyses appear exploratory rather than hypothesis-driven.

• Redundancy exists between text and tables, leading to repetition.

• Clinical relevance of statistically significant findings is not clearly discussed.

• Phrase Non-providers in line 299 –change it to non-providers of tele audiology in practise

Discussion

• The discussion effectively integrates current findings with existing international and regional literature.

• Shift focus from restating results to interpreting their significance.

Conclusion

• The conclusion succinctly summarises the main findings and reinforces the relevance

• The conclusion introduces ideas (e.g., equity and sustainability) not fully developed earlier. Align conclusions more strictly with the study design and data. Provide 2–3 clearly prioritised, actionable recommendations based on the present study findings.

General Comments:

Remove figures that merely restate tabular data unless they add interpretive value.

Figure 1: The figure title is misleading; it implies comparison by experience, but the figure is referenced in the text in relation to timing of adoption.

Figure 2: Percentages can be clearly labelled on each bar. Consider grouping services into broader categories (e.g., diagnostic vs rehabilitative).

Figure 3: The comparative approach (most vs least effective) adds interpretive value beyond simple frequency counts. The figure does not clarify whether effectiveness is based on clinical outcomes, feasibility, or user satisfaction.

**Do you want your identity to be public for this peer review?** For information about this choice, including consent withdrawal, please see our Privacy Policy

Reviewer #1: No

Reviewer #2: No

---

## [Author Response · Author response to Decision Letter 1]

11 Feb 2026

Reviewer #1: The paper could have had very valid information if published a year or two ago. There is currently sufficient evidence on telepractice-related issues. Many are using it as a routine. Since the authors have focused on the arab region, it can still be considered. Though the author declared data available, I could not get access. The following points need attention

Thank you for the feedback. While there is a global transition to teleaudiology, information from Arab countries is limited to a few studies, and services provided are generally discussed in previous studies without detailing which services are currently used. The aim of this study was to shed light on the services currently used and the challenges faced by audiologists.

Data are fully available on

https://zenodo.org/uploads/17344611

1. Aims need to be rewritten. The aims listed sound more like objectives rather than aims

The aims and objectives section was edited accordingly. The objectives were clearly labelled and were as follows:

• To examine the types of audiology services that are delivered remotely by practitioners.

• To understand audiologists’ attitudes towards and perceptions of teleaudiology.

• To explore the challenges faced by audiologists when providing teleaudiology services.

• To gain insight into the perspectives of audiologists who do not currently offer teleaudiology services.

2. It is good to give stats on which countries and their percentage rather than just mentioning two countries and others are less than 10%. Because if there are only one or two from other countries, it is good to exclude those data and re-title to the region focused.

Stats for the audiologists from different countries were added to the description of the audiologists in the Methods section

3. The outcomes of the study highlighted, though not unique, but author can highlight with emphasis to the region, focusing on what aspects can be better.

The Outcomes have been edited according to the comment, and recommendations were added.

Reviewer #2: Reviewer's comment:

Reviewer #2: Abstract:

• The abstract is structured and clearly reports background, methods, key results, and conclusions, allowing readers to quickly understand the scope of the study.

Thank you for this positive comment.

• Temper the conclusion by explicitly stating that findings reflect perceptions rather than effectiveness or outcomes.

Thank you for your comment. The conclusion paragraph in the abstract section has been revised to reflect audiologists' perceptions rather than the treatment's effectiveness.

Reviewer #2: Introduction

• The introduction provides a comprehensive overview of teleaudiology, supported by relevant and up-to-date international literature.

Thank you for this positive comment.

• The introduction is overly long and reads more like a narrative review than a focused rationale for the present study.

The introduction section was shortened to better reflect the study's aims.

Repetition is evident when discussing COVID-19–related acceleration of telehealth adoption. Methodological limitations of previous studies are not sufficiently highlighted. Add critical comparisons between studies rather than descriptive summaries.

Care was taken to eliminate the repetition related to COVID-19. Furthermore, the methodological limitations of previous studies were highlighted. Critical analysis of the studies was performed

• The transition from global literature to the Arab-region context is gradual and lacks a strong problem statement. Add a concise paragraph clearly identifying the research gap and problem statement.

The structure of the introduction was revised to present the global literature and the transition to the Arab region, and a final paragraph before the aims and objectives was added to identify the research gap and the problem statement.

• Research questions or hypotheses are not explicitly articulated.

The research questions were added at the end of the Introduction before the Aims and objectives section.

Methods

• Ethical approval, participant consent, and statistical procedures are clearly documented, enhancing transparency and ethical rigor.

Thank you for the positive comment

• No validation process for the adapted questionnaire is described beyond internal consistency. Describe content or face validity procedures for the survey also the content validity Index (CVI) for the questionnaire used.

Thank you for this valuable comment. The survey was adapted and curated based on a comprehensive review of previously published telehealth and teleaudiology instruments. Items were mapped to established frameworks assessing telehealth readiness, clinician attitudes, service feasibility, perceived barriers, and satisfaction. Content and face validity were established through expert review by five senior audiologists with experience in telepractice, ensuring relevance, clarity, and contextual appropriateness for the Middle East.

• The rationale for grouping providers vs. non-providers is not explained in the study.

Thank you for the comment. The rationale for grouping the providers and non-providers was addressed in the methodology.

“While the focus of the study was Audiologists who used teleaudiology the survey also included questions for audiologists who did not have past experience with teleaudiology to explore their perception of teleaudiology.”

• Country-wise sample imbalance (e.g., dominance of Jordan) is not addressed.

Thank you for the comment. In the sample size section of the methodology, it was mentioned that only a few countries have educational programs in Audiology, with Jordan having the oldest program.

The imbalance among countries is due to the lack of audiology educational programs in many Arab countries, with Jordan having the oldest undergraduate program in the region.

• Potential response bias due to online distribution of the questionnaire is not discussed.

The response bias was addressed in the Limitation and future direction section “However, as with all surveys, there are limitations regarding the reach and representativeness of the sample, creating response bias that limits responses to individuals familiar with digital platforms.”

Results

• Results are detailed, systematically presented, and supported by tables and figures that enhance clarity.

Thank you. The results section was edited to preserve the content and summarise the findings while eliminating repetition.

• The results section is excessively long and contains interpretative statements better suited for the discussion.

Thank you for this comment. All interpretative statements were removed from the results section and addressed in the Discussion section

• Some statistical findings (e.g., extremely small odds ratios) are reported without sufficient explanation.

We acknowledge this concern. The Results section has been revised to report odds ratios descriptively without interpretation. Small odds ratio were interpreted in the discussion

• Multiple analyses appear exploratory rather than hypothesis-driven.

We agree. These analyses were exploratory in nature. This has now been explicitly stated in the Methods and Discussion sections, and the Results section has been revised to present findings descriptively without implying confirmatory hypotheses.

• Redundancy exists between text and tables, leading to repetition.

Thank you for the comment. The redundancy has been addressed and information repeated were removed from the text.

• Clinical relevance of statistically significant findings is not clearly discussed.

Thank you for the comment. The discussion section was updated to address the clinical significance of the findings.

• Phrase Non-providers in line 299 –change it to non-providers of tele audiology in practise

The phrase has been changed throughout the manuscript.

Discussion

• The discussion effectively integrates current findings with existing international and regional literature.

Thank you for the positive comment.

• Shift focus from restating results to interpreting their significance.

Thank you for the feedback. The discussion has been edited by removing the results statements and reporting the significance of the findings

Conclusion

• The conclusion succinctly summarises the main findings and reinforces the relevance

Thank you for the positive comment.

• The conclusion introduces ideas (e.g., equity and sustainability) not fully developed earlier. Align conclusions more strictly with the study design and data. Provide 2–3 clearly prioritised, actionable recommendations based on the present study findings.

The conclusion section has been aligned to the studies design and data, equity and sustainability has been removed.

General Comments:

Remove figures that merely restate tabular data unless they add interpretive value.

Figure 1: The figure title is misleading; it implies comparison by experience, but the figure is referenced in the text in relation to timing of adoption.

The figure's legend was edited for accuracy.

Figure 2: Percentages can be clearly labelled on each bar. Consider grouping services into broader categories (e.g., diagnostic vs rehabilitative).

The percentages were added to the bars, and the services were grouped into broader categories

Figure 3: The comparative approach (most vs least effective) adds interpretive value beyond simple frequency counts. The figure does not clarify whether effectiveness is based on clinical outcomes, feasibility, or user satisfaction.

The legend of this figure has been edited to reflect the perspective of the clinicians

---

## [Editor Report · Decision Letter 1]

12 Feb 2026

Adoption and Implementation of Teleaudiology as a Telehealth Model in Jordan and Arab Countries: A Cross-Sectional Survey

PONE-D-25-56541R1

Dear Dr. AlOmari,

We’re pleased to inform you that your manuscript has been judged scientifically suitable for publication and will be formally accepted for publication once it meets all outstanding technical requirements.

Kind regards,

Rohit Ravi, Ph.D.

Academic Editor

PLOS One

Additional Editor Comments (optional):

Dear Authors, the revision is satisfactory.
---

## [Editor Report · Acceptance letter]

PONE-D-25-56541R1

PLOS One

Dear Dr. AlOmari,

I'm pleased to inform you that your manuscript has been deemed suitable for publication in PLOS One. Congratulations! Your manuscript is now being handed over to our production team.

Kind regards,

on behalf of

Dr. Rohit Ravi

Academic Editor

PLOS One